# The First Cretaceous Epyrine Wasp (Hymenoptera: Bethylidae): A New Genus and Species from Early Cenomanian Kachin Amber

**DOI:** 10.3390/insects15050318

**Published:** 2024-04-30

**Authors:** Manuel Brazidec, Volker Lohrmann, Vincent Perrichot

**Affiliations:** 1CNRS, Géosciences Rennes, University Rennes, UMR 6118, 35000 Rennes, France; manuel.brazidec@gmail.com; 2Institut de Systématique, Évolution, Biodiversité (ISYEB), Muséum national d’Histoire naturelle, CNRS, Sorbonne Université, EPHE, Université des Antilles, CP50, 57 rue Cuvier, F-75005 Paris, France; 3State Key Laboratory of Palaeobiology and Stratigraphy, Nanjing Institute of Geology and Palaeontology and Center for Excellence in Life and Paleoenvironment, Chinese Academy of Sciences, 39 East Beijing Road, Nanjing 210008, China; 4Übersee-Museum Bremen, Bahnhofsplatz 13, 28195 Bremen, Germany; v.lohrmann@uebersee-museum.de

**Keywords:** Hymenoptera, Chrysidoidea, fossil record, taxonomy, Cenomanian, Myanmar

## Abstract

**Simple Summary:**

The Bethylidae, dubbed “flat wasps” because of the general aspect of their bodies, are important components of past and present entomofauna. Currently comprising 3000 species, the Bethylidae are also very diverse in the fossil record. In this study, we present the first Cretaceous record of the Epyrinae, named *Hukawngepyris setosus*, from the rich mid-Cretaceous (*ca.* 100 Ma) Myanmar amber. The newly proposed binomial name is a reference to the Hukawng Valley, where the specimen was collected, and to the conspicuous setation of the species. Describing this fossil adds valuable insights to our understanding of Cretaceous Bethylidae, as it represents the oldest record of the second largest subfamily, which was very common during the Cenozoic. This hints at the need for further exploration of various Cretaceous deposits to enhance our understanding of their past distribution and early occurrences.

**Abstract:**

The Epyrinae are the second largest subfamily of Bethylidae and the most diverse in the fossil record. However, although six of the nine bethylid subfamilies are known during the Cretaceous (either as compression or amber fossils), the Epyrinae were hitherto unknown before the lower Eocene. In this contribution, we report the discovery of the oldest member of this group, based on a female specimen from the early Cenomanian amber of Kachin, Myanmar. We describe and illustrate a new genus and species, *Hukawngepyris setosus* gen. et sp. nov. The new genus is compared with the other epyrine genera and characterized by a unique combination of characters not known in the subfamily. *Hukawngepyris setosus* gen. et sp. nov. is especially unique in the configuration of the forewing venation, with a complete 2r-rs&Rs vein, curved towards the anterior wing margin, and the presence of three proximal and three distal hamuli. The key to the genera of Epyrinae is accommodated to include the newly erected genus.

## 1. Introduction

The Bethylidae (Chrysidoidea) are a diverse and cosmopolitan family of aculeate wasps, with around 3000 extant species [1] and more than 100 fossil species (see summary: [2]; and other recent discoveries: [3,4,5,6,7,8]). As far as is known, all species for which the biology has been reported are parasitoids of lepidopteran and coleopteran larvae [9]. Currently, nine subfamilies are recognized for the Bethylidae, i.e., the extant Bethylinae, Epyrinae, Mesitiinae, Pristocerinae, and Scleroderminae and the extinct Elektroepyrinae, Lancepyrinae, Protopristocerinae, and Holopsenellinae [10]. The Holopsenellinae, however, are not considered a bethylid subfamily any longer since its type genus, i.e., *Holopsenella*, has been transferred into its own family (as Aculeata incertae sedis) [11], leaving the remaining three genera formerly included in Holopsenellinae as orphans until the erection of a new subfamily for them, i.e., the Cretabythinae [8]. The family dates back to the Lower Cretaceous [12] and has a well-documented geological record with fossil representatives known for all but one subfamily, albeit with a larger proportion of fossils within the Epyrinae [13].

The Epyrinae comprise 19 genera and about 1000 species [14] and are considered the second most species-rich subfamily of Bethylidae, outnumbered only by the Pristocerinae [1]. Epyrine representatives are common in upper Eocene Baltic and Rovno ambers, with 15 species recorded [13], and two species are known in Miocene Dominican and Mexican ambers [3]. The oldest epyrine wasps have been described from the lower Eocene amber of France [15,16]. However, while the Bethylidae are quite frequent in Upper Cretaceous ambers (although understudied; [7,13,17,18,19], the Epyrinae were hitherto unknown from this period. The early Cenomanian amber from the Kachin State, Northern Myanmar, is one of the richest fossiliferous insect deposits in the world and a promising source to explore the diversity of fossil Bethylidae [20]. The aim of the present study is to describe and illustrate the first Cretaceous Epyrinae preserved in a piece of mid-Cretaceous Kachin amber from Myanmar.

## 2. Materials and Methods

The study is based on a single specimen preserved in an amber piece that was collected from the deposit of Noije Bum, in the Hukawng Valley, Kachin State, Northern Myanmar [21]. An Early Cenomanian age (98.79 ± 0.62 Ma) was suggested for Kachin amber, based on radiometric data of zircons from volcanic clasts found within the amber-bearing sediments [22] and taphonomic analysis of some amber pieces [23]. Some ammonites found in the amber-bearing bed and within amber corroborate a ate Albian–early Cenomanian age [24,25].

The amber piece is deposited in the amber collection of the Museum für Naturkunde, Berlin, Germany (MB.I.8638). It originates from the private collection of Jens-Wilhelm Janzen and was part of a collection of aculeate Hymenoptera kindly made available to one of us (V. Lohrmann) in 2013 and that was subsequently purchased by the Museum in 2018.

The amber piece has been ground and polished to optimize the observation of the specimen under different views, using wet silicon carbide papers on a grinder polisher (Buehler MetaServ 3000). Photographs were conducted with a Leica DMC4500 camera attached to a Leica M205C stereomicroscope. All images are digitally stacked photo-micrographic composites of several focal planes, which were obtained using Helicon Focus 6.7. Adobe Illustrator CC2019 and Photoshop CC2019 software were used to compose the figures and ImageJ 1.53 for measurements [26]. The morphological examination of the specimen has been carried out using a Leica MZ APO and a Leica M205C stereomicroscope. The description of the characters largely follows the nomenclature of Lanes et al. [27] with the exception of the forewing venation, which follows Azevedo et al. [1]. Additionally, we prefer to use “metasomal segments” (with respective numbers) over “abdominal segments” and fore, mid, and hind legs over pro-, meso- and metalegs. The description of surface sculpturing follows Harris [28]. Main measurements and indices used are as follows throughout the text and figures: LFW = length of forewing; LH = length of head, measured from vertex to apex of clypeus; WH = width of head above eyes; WF = width of frons; HE = height of eye; OOL = ocello-ocular line, the shortest distance between posterior margin of eye to the anterior margin of the anterior ocellus; WOT = width of ocellar triangle, the shortest distance between posterior ocelli; DAO = diameter of anterior ocellus; VOL = vertex-ocular line, the shortest distance between posterior margin of eye to the vertex crest (Figure 1).

This published work and its new nomenclatural acts are registered in ZooBank with the following LSID (reference): https://zoobank.org/6EE95122-16FD-4359-A11A-D53E2EAB6C2E (accessed on 25 April 2024).

## 3. Systematic Paleontology

**Superfamily** Chrysidoidea Latreille, 1802

**Family** Bethylidae Haliday, 1839

**Subfamily** Epyrinae Kieffer, 1914

**Genus** *Hukawngepyris* gen. nov.

https://zoobank.org/5E418042-63B6-4688-91C9-4E2D6D9D0465 (accessed on 25 April 2024).

**Etymology.** The name is a combination of Hukawng, as a reference to the Hukawng Valley where Kachin amber is widely found, and *Epyris*, the type genus of the Epyrinae. Gender masculine.

**Type species.** *Hukawngepyris setosus* sp. nov., by monotypy.

**Diagnosis.** Females of the new genus stand out of all known bethylid genera by the following combination of characters: Vertex, gena, pronotum, mesopleuron, and mesoscutellum foveolate; head and anterior mesosoma with scattered long erect setae (Figure 2 and Figure 3A); compound eyes without setation, 0.6 × HL; malar space convergent, longer than VOL; median clypeal lobe triangular, projecting; mandible elongate, quadridentate; antenna filiform (Figure 3B); scape and pedicel bearing long thick setae; pedicel shorter than each of the flagellomeres; anterior ocellus crossing supra-ocular line; distance between anterior and posterior ocelli shorter than width of ocellar triangle; occipital carina complete. Pronotum longer than wide, with anterior angle rounded; without longitudinal sulcus (Figure 3B); notaulus and parapsidal signum present; sulcus of the mesoscuto-mesoscutellar suture present; mesoscutellum not contacting metapectal–propodeal complex (Figure 3C); metapostnotal median carina complete, straight, extending on propodeal declivity (Figure 4A). Forewing with C, Sc + R, M + Cu, A, Rs&M, cu-a, and 2r-rs&Rs tubular; 2r-rs&Rs tubular and curved, distally reaching anterior wing margin. Hind wing with three straight proximal and three hook-like distal hamuli (Figure 2 and Figure 4C). Protibial spur long and stout (Figure 3C); tarsal claw bifid (Figure 4B). Metasomal tergum 2 entirely smooth (Figure 4B).

*Hukawngepyris setosus* sp. nov.

https://zoobank.org/676B52A5-2D79-433C-B379-0D15A40BC31C (accessed on 25 April 2024).

**Material.** Holotype number MB.I.8638, a nearly complete female, with right side of the first and second metasomal terga damaged and parts of the right midleg and right fore and hind wings missing.

**Etymology.** This species epithet refers to the conspicuous setation of the new species. The species epithet is to be treated as an adjective.

**Diagnosis.** As for genus.

**Type locality and age.** Noije Bum, Hukawng Valley, Kachin State, Northern Myanmar; late Albian–early Cenomanian, ca. 99 Ma, mid-Cretaceous.

**Description.** As for the genus diagnosis, with the following emendation: Body stout, length 4.39 mm, with some metallic reflections. Head rugulose; anteromesoscutum minutely rugulose; metasoma smooth. Head and anterior mesosoma with setae longer than the distance between neighboring setae.

**Head** prognathous, ovoid in lateral view, slightly longer than wide; LH: 0.58 mm, WH: 0.46 mm; WF: 0.34 mm, HE: 0.31 mm; OOL: 0.19 mm; WOT: 0.18 mm; DAO: 0.06 mm; VOL: 0.13 mm; frons convex in lateral view; compound eye elliptical, protruding, located laterally on head, covering 0.6 × LH; median clypeal lobe slightly projecting forward, lateral lobe reduced, median clypeal carina not particularly developed, not continuing on frons; mandible 3 × longer than basal width, 4 short teeth of similar size; palpal formula 6:3; antenna inserted close to median clypeal carina; scape cylindrical, length 0.24 mm, 1.7 × longer than pedicel; pedicel pyriform, length 0.14 mm, base narrower than apex, shorter than flagellomere 1; flagellum with 11 flagellomeres, flagellomeres elongate, cylindrical, distinctly longer than wide (length 0.10–0.17 mm; width 0.04–0.05 mm), no conspicuous setation; flagellomere 11 longest, tapering at apex; ocellar triangle posteriad on head.

**Mesosoma** shorter than metasoma, length 1.12 mm; propleuron barely visible in dorsal view; prosternum reduced; dorsal pronotal area slender (length 0.28 mm), without carina, lateral pronotal margins slightly incurved, posterior margin almost straight; anteromesoscutum slightly shorter than dorsal pronotal area (length 0.24 mm), notauli deeply impressed, converging posterad, posteriorly separated by width of sulcus of mesoscuto-mesoscutellar suture; sulcus of mesoscuto-mesoscutellar suture reniform; metanotum almost not visible; metapectal–propodeal complex wider than long, dorsally carinate, metapostnotal-propodeal suture present, short; propodeal declivity concave, long. **Forewing** developed (LFW: 1.62 mm), hyaline, homogeneously micro-pubescent; pterostigma elongate, 3.9 × longer than wide, narrow; 2r-rs&Rs originating from distal half of pterostigma and widely arched; costal [C], radial [R] and cubital [Cu] cells closed; 1Rs slightly angled at junction with 1M (both veins of similar length); 1M evenly incurved; 1Rs straight; no other tubular veins present but patterns of flexion lines visible in the distal section of the wing. **Hind wing** hyaline, homogeneously micro-pubescent, 1.01 mm long; Sc + R and A veins tubular, Sc + R reaching half the distance to distal hamuli, anal vein length about 0.25 × length of clavus of hind wing. **Legs** particularly pubescent, slender; tibial spurs formula 1–1–2; protibial spur with bifurcate apex; tarsal claws slightly curved, at least metatarsal claw with one subapical tooth.

**Metasoma** smooth, fusiform, length 1.23 mm (indicative), posterior segments partly retracted; six terga visible; petiole short; length of measurable terga: T1 = 0.24 mm; T2 = 0.35 mm; T3 = 0.15); metasomal sterna with few isolated erect bristles; apex of sting exserted.

## 4. Key to the Genera of Epyrinae (Modified from Colombo et al., [14])

**1.** Mesoscuto-mesoscutellar suture with foveae connected by sulcus…………….…….… 2

**-** Mesoscuto-mesoscutellar suture with foveae not connected by sulcus………………..... 10

**2.** Eye densely setose………….………………………………………………...……. *Anisepyris*

**-** Eye glabrous or at most slightly setose………………………….…………………………… 3

**3.** Dorsal pronotal area much wider than long………….…………………...……. †*Gloxinius*

**-** Dorsal pronotal area as long as or longer than wider…………………………………….... 4

**4.** Mesoscuto-mesoscutellar sulcus usually smooth, posterior margin incurved; clypeus with inconspicuous lateral lobes……………………………………………….………………. 5

**-** Mesoscuto-mesoscutellar sulcus slightly scrobiculate, posterior margin straight; clypeus with conspicuous lateral lobes………………………………………...……………………….. 8

**5.** 2r-rs&Rs vein curved towards and reaching anterior wing margin.……..†*Hukawngepyris*

**-** 2r-rs&Rs vein curved towards wing tip, not reaching margin or absent…………...….… 6

**6.** Body and wings with thick black setae……………………………………………… *Laelius*

**-** Body and wings with thin light castaneous setae………………………………………….. 7

**7.** Forewing with 2r-rs&Rs vein absent or faintly visible; protarsus and mesotibiae with spine-shaped setae (♀); genitalia with cuspis undivided (♂)……………………... *Austrepyris*

**-** Forewing with developed 2r-rs&Rs; protarsus and mesotibiae without spine-shaped setae (♀); genitalia with cuspis divided (♂)………….………………………………. *Chlorepyris*

**8.** Eye medium-sized, occupying at most 2/3 of side of head; median clypeal lobe as long as lateral ones or reduced; dorsal pronotal area usually carinate………..….……... *Rysepyris*

**-** Eye remarkably large, occupying almost all sides of the head; median clypeal lobe shorter than lateral ones; dorsal pronotal area usually ecarinate……………...…………………….. 9

**9.** Protarsus with long spatulate spine-shaped filament (♀); forewing with 2r-rs&Rs vein short; anteromesoscutum without foveolate transverse groove; posterior corner of metapectal-propodeal complex one-toothed……………………………………………. *Disepyris*

**-** Protarsus without spatulate spine-shaped filament (♀); forewing with 2r-rs&Rs vein long; anteromesoscutum with foveolate transverse groove; posterior corner of metapectal-propodeal complex two-toothed…………………………..…………………………. *Holepyris*

**10.** Distance between posterior ocelli much less than their own diameter……... *Trissepyris*

**-** Distance between posterior ocelli more than their own diameter………………………. 11

**11.** Dorsal pronotal area with wider anterior elevation…….……. *Aspidepyris* (type pattern)

**-** Dorsal pronotal area without such elevation………………………..……………………... 12

**12.** Dorsal pronotal area anterior margin incurved; humeral angle angulate…... *Calyozina*

**-** Dorsal pronotal area anterior margin straight or outcurved; humeral angle rounded... 13

**13.** Anterior area of the S2 with flap scale-shaped (♂♀); antenna usually pectinate; flagellomere 1 very reduced (♂)………………………….……………………………………. *Calyoza*

**-** Anterior area of the S2 without flap; antenna filiform; flagellomere 1 not reduced…... 14

**14.** Foveae of mesoscuto-mesoscutellar suture distant by less than their half own….….. 15

- Foveae of mesoscuto-mesoscutellar suture distant by more than their half own….…... 17

**15.** Metapectal-propodeal disc with indistinct metapostnotal carinae…………... *Idatepyris*

**-** Metapectal-propodeal disc with distinct metapostnotal carinae………………………... 16

**16.** Dorsal pronotal area usually carinate (♂♀); metapostnotal-propodeal carina sinuous (♀); apical aedeagal lobe with conspicuous projection ventrad (♂)………………. *Bakeriella*

**-** Dorsal pronotal area ecarinate (♂♀); metapostnotal-propodeal carina never sinuous (♀); apical aedeagal lobe without such projection (♂)……………………………………….. *Epyris*

**17.** Mesoscuto-mesoscutellar foveae distant by less than 4 × own width………………... 18

**-** Mesoscuto-mesoscutellar foveae distant by more than 4× own width…………………. 19

**18.** Mesoscuto-mesoscutellar fovea drop-shaped; cuspis undivided…………….. *Psilepyris*

**-** Mesoscuto-mesoscutellar fovea circular/oval-shaped; cuspis divided partially……. *Dolus*

**19.** Antennal scape with spine pecten-shaped; ventral margin of mandible with laminar expansion (♀); cuspis and harpe very long (♂)……………………………………. *Muellerella*

**-** Antennal scape without spine pecten-shaped; ventral margin of mandible without such expansion; cuspis and harpe short (♂)…………………………………………….…………. 20

**20.** Eye flat; notaulus faint (♂♀); dorsal mandibular tooth curved dorsad (♂)….. *Gracilepyris*

**-** Eye protuberant; notaulus well-impressed (♂♀); mandible with dorsal tooth mesad (♂) …………………………………………………………………… *Aspidepyris* (non-type pattern)

## 5. Discussion

Four of the nine subfamilies currently assigned to the Bethylidae are exclusively known from the fossil record but *Hukawngepyris* gen. nov. does not fit in any of them. The new fossil can be distinguished from the Cretabythinae by displaying a reduced forewing venation with only three cells fully surrounded by tubular veins and the metanotum much obscured by the mesoscutellum [29,30]; from the Lancepyrinae by the absence of the forewing vein Rs + M, and the pronotum not anteriorly strongly narrowing [8]; from the Protopristocerinae by the metanotum much obscured by the mesoscutellum [30]; and from the Elektroepyrinae by the absence of the forewing vein 3Cu [10,15].

Of the five extant subfamilies, three have a fossil record that dates back to the mid-Cretaceous, i.e., the Pristocerinae [31], the Scleroderminae [32,33], and the Bethylinae [1,34]. Whereas the first two are known from specimens included in Kachin amber (although Cockerell’s attribution of the Scleroderminae specimen to this subfamily is difficult to rely on [33]), the Bethylinae are known from the Lower Cretaceous of Transbaikala (Russia). Females of the Pristocerinae are apterous, even in the fossil record, excluding this subfamily [1,31]. The Scleroderminae are generally characterized by a polished cuticle, a reduced forewing venation with C vein often absent (most complete venation found in Nothepyris Evans, 1973), the anterior wing margin angularly incurved, and the pedicel longer than the first flagellomere [1,35], precluding an assignment of our specimen to this subfamily. Our specimen displays some characteristics reminiscent of the Bethylinae, namely the 2r-rs&Rs vein reaching the anterior margin of the wing and the somewhat angled 1Rs&1M vein. However, it lacks the tubular Rs + M vein, the clypeal carina extending on the frons, and the angled tarsal claws, diagnostic for this subfamily [1]. The only bethylid subfamily that has no documented fossil record at all are the Mesitiinae, members of which have a conspicuously long second metasomal segment that is longer than all subsequent segments. The metasoma of our specimen superficially resembles that of Mesitiinae as it is contracted, with the posterior segments retracted and not entirely visible, giving the illusion that the metasomal segment 2 is much longer than the subsequent segments. This contraction, however, may be due to a muscle contraction during its embedment in amber or death, so it is likely that the metasomal segment 2 is actually not longer than the posterior segments at all.

We attribute our specimen to the Epyrinae, as it shows more affinities with this subfamily than with any of the other bethylid subfamilies. The Epyrinae are challenging to define based on their morphology due to their large phenotypic plasticity [36]. Alencar and Azevedo [37] proposed a diagnosis for the subfamily they just re-established, later followed by Azevedo et al. [1]. Among the diagnostic characters, our specimen displays the antennae with 11 flagellomeres, the 6:3 palpal formula, the clypeus divided in three lobes with the lateral lobe reduced (present in most genera but not all), the anteromesoscutum with notauli, the metapectal–propodeal complex dorsally carinate with metapostnotal median carina complete, the forewing developed with only three closed cells. These characters are more or less shared with other bethylid subfamilies but Alencar and Azevedo [37] additionally identified three synapomorphies: the propleural epicoxal sulcus circular, the anterior area of the sternite 1 (=petiolar root) outcurved with medial emargination, and fused to the posterior area of the sternite 1 (=body root). In our specimen the propleural epicoxal sulcus is circular (Figure 3C) rather than elongate [37]. The configuration of the sternite 1 is difficult to assess on the specimen but based on the characters described above, it is likely that it belongs to the Epyrinae.

Within the Epyrinae, following the key of Colombo et al. [14], our specimen keys in *Chlorepyris* Kieffer, 1913 because of the following couplets: (1) mesoscuto-mesoscutellar suture with foveae connected by sulcus; (2) eye glabrous; (3) dorsal pronotal area distinctly longer than wide; (4) sulcus of the mesoscuto-mesoscutellar suture incurved, clypeus without lateral lobe; (5) body without thick black setae (see below for details on that character); (6) 2r-rs&Rs tubular, pigmented. However, it differs from that genus in having quadridentate mandibles. On the basis of the Colombo et al. key, the other epyrine genera are excluded as follows: *Trissepyris* Kieffer, 1905, *Aspidepyris* Evans, 1964, *Caliozyna* Enderlein, 1912, *Calyoza* Hope, 1837, *Idatepyris* Colombo, Tribull and Azevedo, 2022, *Bakeriella* Kieffer, 1910, *Epyris* Westwood, 1832, *Psilepyris* Kieffer, 1913, *Dolus* Motschulsky, 1863, *Muellerella* Saussure, 1892 and *Gracilepyris* Colombo, Tribull and Azevedo, 2022 have the mesoscuto-mesoscutellar suture with foveae not connected by a sulcus (couplet 1); *Anysepyris* Kieffer, 1905 has strongly setose eyes (couplet 2); the extinct *Gloxinius* Colombo and Azevedo, 2021 has the dorsal pronotal area wider than long (couplet 3); *Rysepyris* Kieffer, 1906, *Disepyris* Kieffer, 1905 and *Holepyris* Kieffer, 1904 have the sulcus of the mesoscuto-mesocutellar suture straight and a large lateral clypeal lobe (couplet 4); and *Austrepyris* Colombo, Tribull and Azevedo, 2022 has the 2r-rs&Rs vein spectral (couplet 6). *Laelius* Ashmead, 1893 is mainly characterized by possessing thick long black setae on the body and forewing [38] and our specimen displays some long erect setae, superficially similar to that of the aforementioned genus. Actually, *Laelius* have sparser and longer setae, sometimes being as long as HE, a shorter pronotum, and the 2r-rs&Rs vein projected straightly toward the wing tip (when present) (couplet 5). Therefore, we propose the new genus *Hukawngepyris* gen. nov. to accommodate this specimen. This genus is composed of only one species, *Hukawngepyris setosus* sp. nov. *Hukawngepyris setosus* gen. et sp. nov. is particular within the Epyrinae for possessing the following combination of characters: long setae scattered on the body, the 2r-rs&Rs vein fully pigmented, curved towards the anterior wing margin, the 1Rs&1M vein angled, the antennae long, four toothed-mandibles, the mesoscuto-mesoscutellar sulcus present and the metapostnotum with median and lateral carinae present.

A conspicuous characteristic feature of *Hukawngepyris setosus* gen. et sp. nov. is the configuration of the hamuli, with three hamuli grouped proximally and three hamuli grouped distally. The presence of three distal hamuli is common within the Bethylidae (e.g., Scleroderminae: [35]; *Laelius*: [39]; *Afrobethylus* Ramos and Azevedo, 2016: [40]; Mesitiinae: [41]). The presence of proximal hamuli, and moreover in a group of three, is less frequent [42,43], reported only for three species, one in the Mesitiinae, i.e., *Hadesmesitius simplicitus* (Barbosa and Azevedo, 2011), and two in the Pristocerinae, i.e., *Apenesia sahyadrica* Azevedo and Waichert, 2006 [41] and *Pseudisobrachium somaliense* Benoit, 1957 (M. Brazidec pers. obs.). All three species, however, have a different number of distal hamuli compared to *Hukawngepyris setosus* gen. et sp. nov. Therefore, this unique 3:3 configuration of the hamuli is a good additional character to define *Hukawngepyris* gen. nov. as a new genus. The number of hamuli present in *Trissepyris* was briefly discussed, highlighting an asymmetry between both hind wings in the type species and a different number between the two species, leading to the conclusion that it is not a good character for this genus [44]. However, the evolution of the number and position of hamuli in the Bethylidae is unclear and may require a broader study to understand their diagnostic value at the subfamilial and generic levels.

With 32 species, the Epyrinae are the most diverse Bethylidae subfamily in the fossil record, followed by the Bethylinae, mostly known from Cenozoic ambers [45], and the Pristocerinae and the Scleroderminae, of which several occurrences have been reported throughout the Cretaceous [7,18,31]. All but two fossils are attributed to the extant *Epyris*, *Anisepyris*, *Bakeriella*, *Chlorepyris*, *Holepyris*, and *Laelius*; most of them have been described from the upper Eocene Baltic amber [13]. The subfossil Tanzanian copal, Miocene Dominican amber, Miocene Mexican amber, Oligocene Rott Formation, Eocene Bembridge marls, Eocene Rovno amber, Eocene Florissant Formation, and Eocene French amber yielded between one to five species [3,13,16,46,47,48,49,50]. So far, the only extinct epyrine genus was *Gloxinius*, with one species from Baltic amber [51], making *Hukawngepyris setosus* gen. et sp. nov. the second species belonging to a fossil genus. The description of the new species also results in a 47-million-year gap in the Epyrinae fossil record, between the early Cenomanian and early Eocene. Therefore, additional specimens may be discovered after investigation of Upper Cretaceous deposits such as amber from France, Russia, or Canada.

Additionally, this discovery emphasizes a little more on the diversity of the Bethylidae during the mid-Cretaceous. *Hukawngepyris setosus* gen. et sp. nov. represents the earliest known Epyrinae, which is now the sixth bethylid subfamily to be recorded from mid-Cretaceous Burmese ambers (from Hkamti and Kachin), after the Cretabythinae [30], the Protopristocerinae [30], the Lancepyrinae [8], the Pristocerinae [31] and the Scleroderminae [32]. The Elektroepyrinae are known only from Eocene French amber [10] and the Mesitiinae have no fossil record at all. Considering the Epyrinae diversity and their abundance in Cenozoic ambers (most notably from the Eocene), it was surprising that the family had not yet been recorded from the Cretaceous. The discovery of *Hukawngepyris setosus* gen. et sp. nov. pushes back the estimated divergence times of most of the bethylid subfamilies to the Lower Cretaceous and it could be interesting to include this species in a broader phylogenetic context to see how it affects the tree topologies.

## Figures and Tables

**Figure 1 insects-15-00318-f001:**
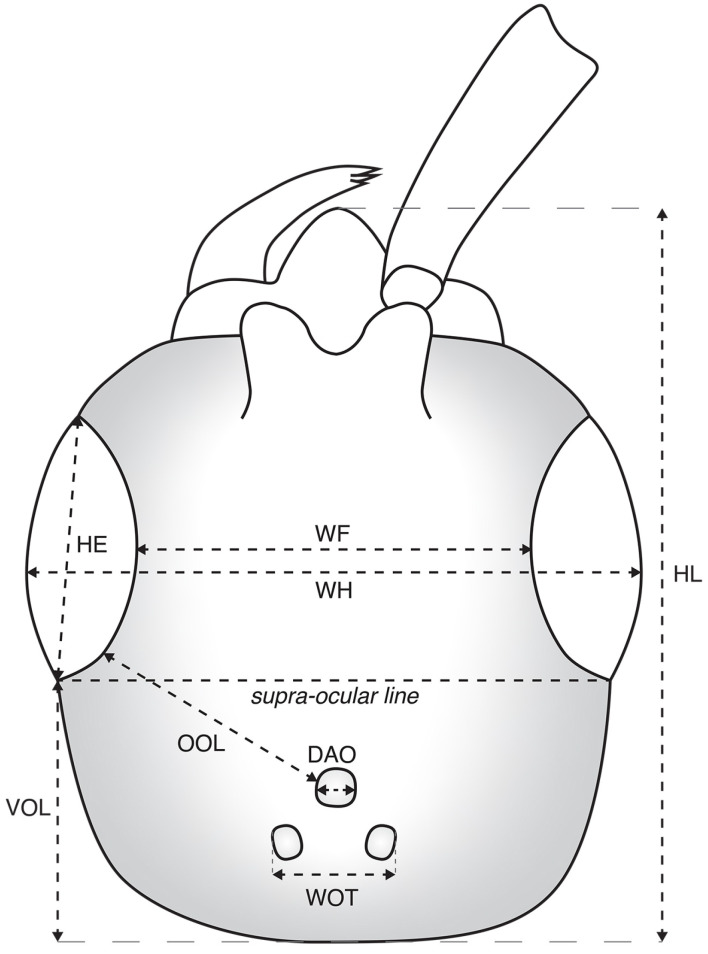
Schematic line drawing of a fictitious bethylid wasp head with indication of measurements used in the present study.

**Figure 2 insects-15-00318-f002:**
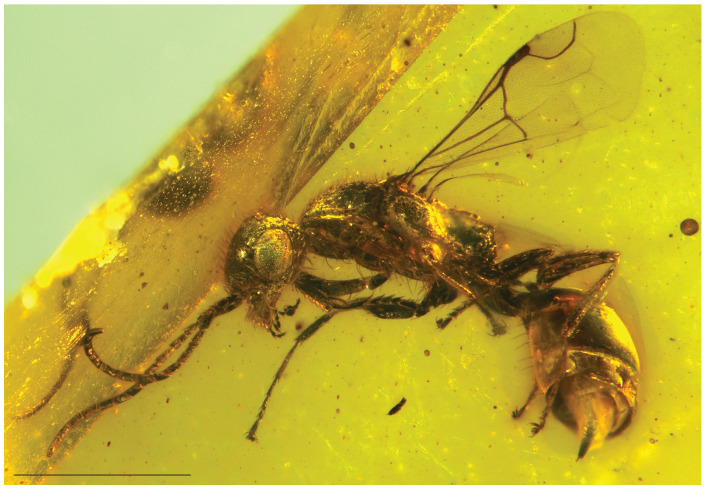
*Hukawngepyris setosus* gen. et sp. nov., female holotype, MB.I.8638, habitus in left lateroventral view. Scale bar = 1 mm.

**Figure 3 insects-15-00318-f003:**
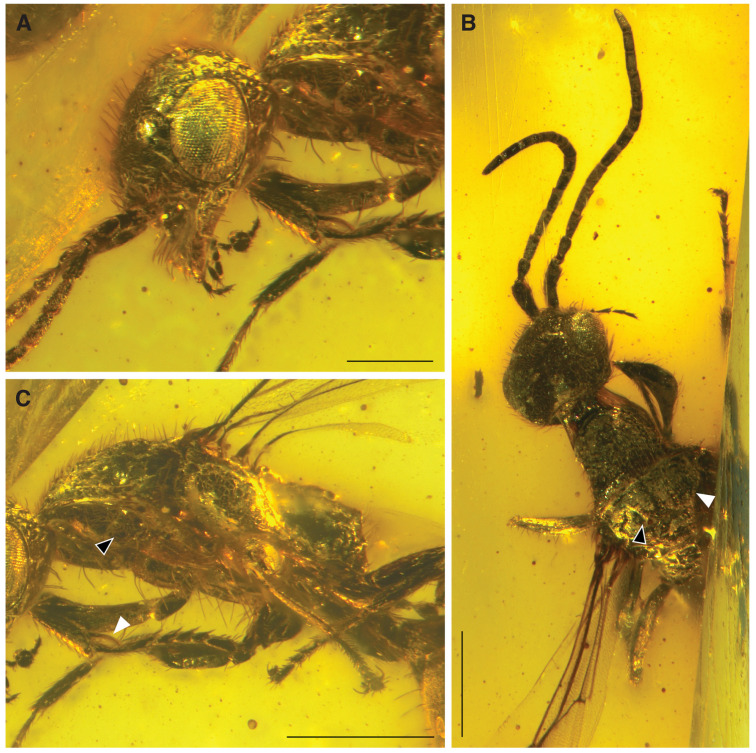
*Hukawngepyris setosus* gen. et sp. nov., female holotype, MB.I.8638, (**A**) head in left lateral view; (**B**) head and partial mesosoma in dorsal view (black arrow: notaulus; white arrow: parapsidal signum); (**C**) mesosoma in left lateroventral view (black arrow: propleural epicoxal sulcus; white arrow: protibial spur). Scale bars: A = 0.25 mm, B–C = 0.5 mm.

**Figure 4 insects-15-00318-f004:**
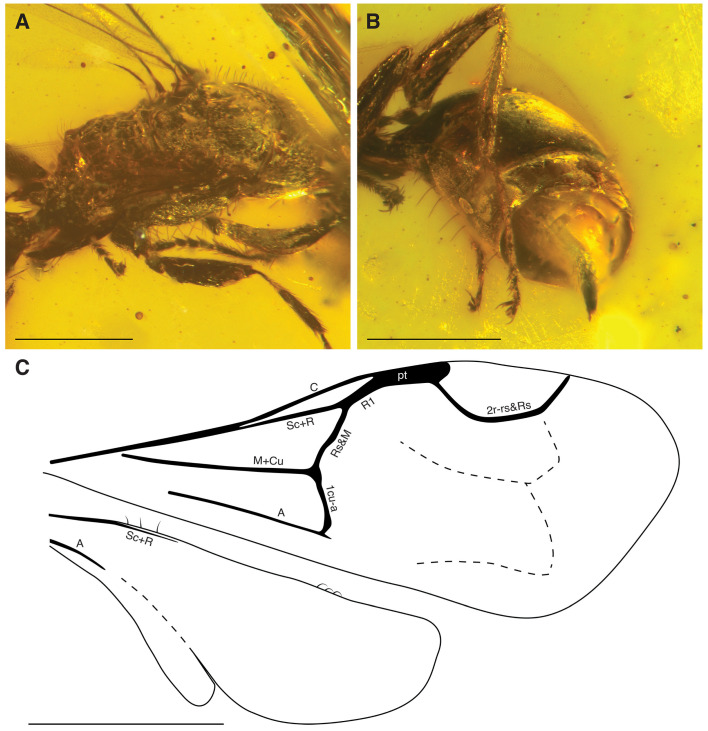
*Hukawngepyris setosus* gen. et sp. nov., female holotype, MB.I.8638, (**A**) mesosoma in dorsolateral view; (**B**) metasoma in left posterolateral view; (**C**) drawing of fore and hind wings (dashed lines indicate flexion lines; pt = pterostigma). Scale bars = 0.5 mm.

## Data Availability

The data presented in this study are available in this article.

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
