# Peer review of "The First Cretaceous Epyrine Wasp (Hymenoptera: Bethylidae): A New Genus and Species from Early Cenomanian Kachin Amber"

_insects, 2024, doi:10.3390/insects15050318_

Round 1

Reviewer 1 Report (Previous Reviewer 2)

Comments and Suggestions for Authors

This manuscript, titled "Theet sp. nov., the first Cretaceous epyrine wasp (Hymenoptera: Bethylidae): a new genus and species from early Cenomanian Kachin amber" makes a significant contribution to the field of palaeoentomology by presenting the description of a new genus and species from the early Cenomanian amber of Kachin, Myanmar. Below, I provide a comprehensive review of the main components of the manuscript.

1- Title: I suspect there is something wrong with the title, as this "Theet sp. nov." Was it supposed to be like that?

2- Abstract: The abstract effectively summarizes the key findings of the study. Please correct "fore wing" to "forewing". 

3- Introduction: The introduction provides essential background information about the biological group.

4- Material and Methods: The methodology section offers a clear and detailed account of the procedures used in the study.

5- Systematic palaeontology:

(a) There seems to be an unnecessary inclusion of the entire key of Epyrinae in this study. Since you only modify point 5 to include the new genus, why not make an addendum to this point? For instance, in Gloxinius, "nov." is introduced (lines 186-187), followed by a formatting error and confusion (lines 188-189). I strongly suggest removing this entire key, as it may not align with your objectives;

(b) I would suggest adding a "remarks" section to incorporate the text from lines 275-319, as it is more suitable for justifying the taxon classification.

4- Discussion: I recommend delving deeper into the implications of the discovery for understanding the paleodiversity of Epyrinae. Overall, the discussion effectively synthesizes the study's results, contextualizes them, and highlights their significance.

5- References: Please review the references. For example, reference 07 (Szabó et al.) has the journal name repeated.

In conclusion, this manuscript provides a well-structured and informative description of a new epyrine genus and species. Its contributions to the field of Bethylidae systematics make it a valuable addition to the scientific literature.

Comments on the Quality of English Language

The English is good and just needs a formatting review and some attention to certain words like 'fore wing', for example.

Author Response

Thank you for the constructive comments and suggestions. Please find our point-by-point response below, where appropriate.

First, we would like to draw your attention on a technical issue during the conversion of the draft from .docx to .pdf by the MDPI online system, which has resulted in numerous formatting errors that were not present in the original .docx manuscript (. As a result, several of the requested changes are pointless. These are listed point by point

1- Title: I suspect there is something wrong with the title, as this "Theet sp. nov." Was it supposed to be like that?

- Reply: Incorrect title due to the technical issue, indeed. The original title has always been: "The first Cretaceous epyrine wasp (Hymenoptera: Bethylidae): a new genus and species from early Cenomanian Kachin amber."

2- Abstract: The abstract effectively summarizes the key findings of the study. Please correct "fore wing" to "forewing". 

- Reply: corrected

5- Systematic palaeontology:

(a) There seems to be an unnecessary inclusion of the entire key of Epyrinae in this study. Since you only modify point 5 to include the new genus, why not make an addendum to this point? For instance, in Gloxinius, "nov." is introduced (lines 186-187), followed by a formatting error and confusion (lines 188-189). I strongly suggest removing this entire key, as it may not align with your objectives;

- Reply: We opted to keep the entirety of the key to the Epyrinae genera. Having it complete will makes it easier to work with for future studies on the subfamilies. This was also suggested by a reviewer during the previous round. 
Regarding some formatting errors, these were again due to the .pdf conversion.

(b) I would suggest adding a "remarks" section to incorporate the text from lines 275-319, as it is more suitable for justifying the taxon classification.

- Reply: Since we discuss the morphology range within the subfamily as well as some diagnostic characters for the new genus, we prefer to keep the systematic remarks in the discussion section.

4- Discussion: I recommend delving deeper into the implications of the discovery for understanding the paleodiversity of Epyrinae. Overall, the discussion effectively synthesizes the study's results, contextualizes them, and highlights their significance.

- Reply: We have added a paragraph summarizing the paleodiversity of the subfamily.

5- References: Please review the references. For example, reference 07 (Szabó et al.) has the journal name repeated.

- Reply: Again, a technical issue from the conversion by the online system. We have checked all references, though. 

Reviewer 2 Report (Previous Reviewer 3)

Comments and Suggestions for Authors

All the key elements are present, namely: title, two kinds of summaries, introduction, methodology, results (named “Systematic paleontology” in the MS), and discussion. The title does not reflect the article contest due to (perhaps) technical error indicating “Theet sp. nov.,” – probably omission in between the letter combination. The Simple Summary reflects the content of the article accurately as also does the abstract. The “Introduction” describes in detail what the authors hoped to achieve accurately, and clearly state the problem being investigated – the absence of any known fossil Epyrinae from Cretaceous up to now. The authors accurately explain how the data was gathered; the design is suitable for answering the question posed. Results are clearly laid out and are presented in a logical sequence. The claim in the “Discussion” section is supported by the results and does seem reasonable. When considering the whole article, the balance of the illustrative material to the text is excellent: eight figures (illustrations and pictures) serve to inform the reader; they are a very important part of the story; they describe the data accurately and they are consistent; in addition, one key to all the seventeen known genera of Epyrinae is presented. The authors make an important contribution to the field of Bethylidae taxonomy with the publication of these results.

Some minor concerns through the text denoted to the line numbers are below:

Line 2. Consider the meaning of “Theet sp. nov.” and change in accordance with the text of the MS.

Line 75. Please, explain the meaning of the abbreviation “VL” here; ? Volker Lohrmann.

Line 107. “nov. Etymology” instead of “nov.Etymology”.

Line 134. “dorsolateral view” instead of “lateroventral view”.

Line 136. “nov. Material” instead of “nov.Material”.

Line 140. “species” instead of “specific”.

Line 186-187. Consider the presence of the “usnov.”

Line 243. Avoid “et sp. nov. nov.”.

Line 367. Recommend to use the full name of the journal/publishing source as follows: Journal of South American Earth Sciences.

Line 378. “Research” instead of “ResearchRes”.

Line 380. Recommend to use the full name of the journal/publishing source as follows: Arthropod Systematics & Phylogeny.

Line 382. Recommend to use the full name of the journal/publishing source as follows: Bulletin of the Museum of Comparative Zoology. Harvard University.

Line 386. Recommend to use the full name of the journal/publishing source as follows: Cretaceous Research.

Line 388. Recommend to use the full name of the journal/publishing source as follows: Current Biology.

Line 394. Recommend to use the full name of the journal/publishing source as follows: Systematic Entomology.

Line 395. Recommend to use the full name of the journal/publishing source as follows: Neues Jahrbuch für Geologie und Paläontologie – Abhandlungen.

Line 402. Recommend to use the full name of the journal/publishing source as follows: Paläontologische Zeitschrift.

Line 404. Recommend to use the full name of the journal/publishing source as follows: Cretaceous Research.

Line 409. Recommend to use the full name of the journal/publishing source as follows: American Museum Novitates.

Line 410. “Shi, G” instead of “J. Res. Shi, G”.

Line 412. Recommend to use the full name of the journal/publishing source as follows: Cretaceous Research.

Line 414. Recommend to use the full name of the journal/publishing source as follows: Earth and Environmental Science Transactions of The Royal Society of Edinburgh.

Line 416. Recommend to use the full name of the journal/publishing source as follows: Journal of Asian Earth Sciences.

Line 419. Recommend to use the full name of the journal/publishing source as follows: Proceedings of the National Academy of Sciences of the United States of America.

Line 420. Recommend to use the full name of the journal/publishing source as follows: Nature Methods.

Line 424. Recommend to use the full name of the journal/publishing source as follows: Occasional Papers in Entomology.

Line 428. Recommend to use the full name of the journal/publishing source as follows: Zoological Journal of the Linnean Society.

Line 429. Recommend to use the full name of the journal/publishing source as follows: Neues Jahrbuch für Geologie und Paläontologie – Abhandlungen.

Line 432. Recommend to use the full name of the journal/publishing source as follows: Historical Biology.

Line 434. Recommend to use the full name of the journal/publishing source as follows: American Journal of Science. Series 4.

Line 436. Recommend to use the full name of the journal/publishing source as follows: Historical Biology.

Line 439-440. Recommend to use the full name of the journal/publishing source as follows: Arthropod Systematics & Phylogeny.

Line 444. Recommend to use the full name of the journal/publishing source as follows: Systematic Entomology.

Line 446. Recommend to use the full name of the journal/publishing source as follows: Zoologischer Anzeiger.

Line 448. Recommend to use the full name of the journal/publishing source as follows: Insect Systematics & Evolution.

Line 452. Recommend to use the full name of the journal/publishing source as follows: Arthropod Systematics & Phylogeny.

Line 456. Recommend to use the full name of the journal/publishing source as follows: Journal of Hymenoptera Research.

Author Response

Thank you for the constructive comments and suggestions. Please find our point-by-point response below, where appropriate.

First, we would like to draw your attention on a technical issue during the conversion of the draft from .docx to .pdf by the MDPI online system, which has resulted in numerous formatting errors that were not present in the original .docx manuscript. As a result, some of the requested changes are pointless. 

Second, we followed all your suggestions and made all of but one the requested corrections that were not related to this technical issue, so we do not address each specifically.
However, the journal style for references requires abbreviated journal names so we did not follow your recommendation to write the full names. 

We hope that this revised manuscript will fulfil your expectations.

This manuscript is a resubmission of an earlier submission. The following is a list of the peer review reports and author responses from that submission.

Round 1

Reviewer 1 Report

Comments and Suggestions for Authors

line 33: key is not provided, rather just one couplet is provided.

line 47: this paper is not focused on the remaining genera of Holopsenellinae, then it is very inappropriate here to make any nomenclatural action of these taxa. Authors did not even described the proposed subfamily. And Boudinot et al. (in press) retrieved the subfamily inside Bethylidae. Thus, here is definitely not adequate to do any nomenclatural change. Besides, a new taxon must have be followed by description, and here requires typification.

line 64: I strongly recommend the book by Lanes eta al. (2020) for terminology. There, authors will see that "posterior corners projected into spines" is called "posterior propodeal projection".

line 105: several mistakes: 1) WOT includes posterior ocelli, here what was called as WOT is indeed POL (posterior ocellar line, that does not include ocelli); 2. WH includes eyes, WH is the maximum head width, wherever it is; 3) VOL is the maximum length between eyes and vertex crest, but always taken parallel not inclined.

line 118: which scutellum? Mesoscutellum or metascutellum?

line 175: "converging posterad" (which means that notauli is convergent in posterior direction) instead "posteriorly converging" (which means that posterior part of notauli is convergent, but still missing the information if it is convergent in anterior or posterior direction). The text is full of this mistake.

line 188: "posterior segments partly retracted" is not a morphological character, instead, that is because the action of any muscle contraction or fossilization collapse, or both.

line 263: Elektromesitius gen. nov. is particular within Mesitiinae for displaying the following combination of characters: dorsal pronotal area without carina, large glabrous eyes covering more than 0.5 × LH, long pubescence on the head and mesosoma, and small posterior propodeal projection. There are several genera of Bethylidae of the other subfamilies with these characters. The section of Discussion of the MS is very focused on differentiating it from the other genera of Mesitiinae, and that is well defended. On the other hand, the fundament of hypothesis of this genus as Mesitiinae is very badly defended. This issue must be completely redraw in order to better allocate the genus. Honestly, I am not able to see a mesitiine here.

Reviewer 2 Report

Comments and Suggestions for Authors

Dear authors,

I have carefully reviewed your manuscript, and I find it to be an interesting contribution to the understanding of the paleofauna of Bethylidae. However, there are some points that require further attention before acceptance, particularly regarding the misidentification of the studied specimen as a Mesitiinae. I have provided below some suggestions and comments to enhance the quality of your paper:

Introduction: a. Holopsenellinae: I recommend including information about all the morphological phylogenies proposed for Bethylidae. Holopsenellinae has been consistently recovered as the sister group of other bethylids, as supported by data published by Boudinot et al. (2022, https://doi.org/10.1101/2022.02.20.480183). b. Diagnostic characters of Mesitiinae: In the description of "metasomal tergite 2 much longer than subsequent ones," consider specifying that T2 is much longer than all subsequent tergites combined. This distinction is crucial, distinguishing it, for example, from Epyrinae and Scleroderminae, which also have a long T2.

Material and Methods: a. WOT: This measurement usually includes the ocelli. Please refer to Evans (1964, p. 10) [A SYNOPSIS OF THE AMERICAN BETHYLIDAE (HYMENOPTERA, ACULEATA)] for clarification.

Results and Discussion:

a. (p.7, lines 199-211): The authors use the taxonomic key from Azevedo et al. (2018) to identify the specimen as a Mesitiinae. However, among the characters mentioned as diagnostic, as well as some new characters present in the specimen, led me to conclude that such a specimen is not a Mesitiinae. First, the metasoma is telescopic, and clearly, in the specimen, it is collapsed; however, it is visible that if it were not, T2 would not be longer than the posterior tergites combined. Regarding the forewings, the specimen has two unique characteristics, not observed until now in Mesitiinae: (1) the 2r-rs&Rs touching the anterior margin (similar to some Bethylinae) and (2) the apparent existence of a vein like a stub in Rs&M (check whether it is a distortion of the amber or if such a vein really exists). Particularly, this specimen is morphologically similar to Holepyris (sensu Colombo et al. 2022), and consequently, it falls into Epyrinae rather than Mesitiinae. b. (p.9, lines 262-274): A brief discussion on the number of hamuli in Bethylidae was proposed by Colombo & Azevedo (2023, https://doi.org/10.5252/zoosystema2023v45a4). c. (p.9, lines 262-274): The combination of characters listed to diagnose the new genus aligns more with Holepyris (Epyrinae), and the authors forgot to list, for example, the characters of the forewings that are truly unique.

I hope these suggestions help improve the clarity and accuracy of your manuscript.

Reviewer 3 Report

Comments and Suggestions for Authors

The article is novel and interesting to publish; it contains a description of a new fossil genus (Elektromesitius) with a new species (hukawngensis). The article does adhere to the journal’s standards in any aspect. The research question is important in terms of understanding the evolutionary history of Mesitiinae in space and time.

Some minor concerns through the text denoted to the line numbers are below:

Lines 50-53. It is worth commenting that the generic affiliation of the newly described species by Theodor Cockerell in 1921 is doubted: “Mesitius (?) rectinervis” on page 21.

Line 81. Please, explain the meaning of the abbreviation “VL” here; ?Volker Lohrmann.

Lines 126-127, 176. The “mesoscuto-mesoscutellar suture” is the correct term here.

Lines 325, 339. Recommend to use the full name of the journal/publishing source as follows: Arthropod Systematics & Phylogeny

Line 326. The reference formatting of the DOI number is wrong; the following is correct: https://doi.org/10.3897/asp.81.e96737

Line 312. Recommend to use the full name of the journal/publishing source as follows: Journal of South American Earth Sciences

Lines 323, 331, 356, 358. Recommend to use the full name of the journal/publishing source as follows: Cretaceous Research

Line 333. The reference formatting of the article title is wrong; the following is correct: Phylogenomic Insights into the Evolution of Stinging Wasps and the Origins of Ants and Bees

Line 333. Recommend to use the full name of the journal/publishing source as follows: Current Biology

Line 335-336. The reference formatting of the article title is wrong; the following is correct: Synopsis of the fossil flat wasps Epyrinae (Hymenoptera, Bethylidae), with description of three new genera and 10 new species

Line 336. Recommend to use the full name of the journal/publishing source as follows: Journal of Systematic Palaeontology

Line 341. Recommend to use the full name of the journal/publishing source as follows: The Annals and Magazine of Natural History, including Zoology, Botany, and Geology. Ninth Series

Line 350. Recommend to use the full name of the journal/publishing source as follows: American Museum Novitates

Line 360. Recommend to use the full name of the journal/publishing source as follows: Earth and Environmental Science Transactions of The Royal Society of Edinburgh

Line 362. Recommend to use the full name of the journal/publishing source as follows: Journal of Asian Earth Sciences

Line 365. Recommend to use the full name of the journal/publishing source as follows: Proceedings of the National Academy of Sciences of the United States of America or the acronym PNAS

Line 366. Recommend to use the full name of the journal/publishing source as follows: Nature Methods

Line 370. Recommend to use the full name of the journal/publishing source as follows: Occasional Papers in Entomology

Line 370. Recommend to use the full name of the journal/publishing source as follows: European Journal of Taxonomy

Author Response

We have made all the corrections requested by the reviewers, so we do not address each specifically here. 

We hope that this revised manuscript will fulfil your expectations.

Reviewer 4 Report

Comments and Suggestions for Authors

The submitted manuscript presents the description of a new genus containing one new fossil species of Mesitiinae, belonging to the family Bethylidae. The novelty is presented as the first fossil record of Mesitiinae. In my opinion, the manuscript is a fine contribution to the knowledge of fossil Bethylidae, and I recommend its publication. However, I have proposed some changes to the text and added some comments to clarify a few pending questions, see attached file. The main question, is the comparison of the new genus with other extinct genera of Bethylidae, even if belonging to other subfamilies. The Authors presented a comparison of the new genus within a subfamilial context, without citing extinct genera, at least the closest relatives. This would be useful for the readers. 

Author Response

(The authors gave the same response as above.)
